# Urban coyotes were observed rarely and retreated consistently from assertive approaches by volunteers in neighborhoods

**Gabrielle Lajeunesse**[1], **Howard W. Harshaw**[2], **Colleen Cassady St. Clair**[1]*

**1** Department of Biological Sciences, University of Alberta, Edmonton, Alberta, Canada, **2** Faculty of Kinesiology, Sport, and Recreation, University of Alberta, Edmonton, Alberta, Canada

* cstclair@ualberta.ca

**Data availability statement:** All data files and R script are publicly available on Figshare, and can be accessed from the following link: https://doi.org/10.6084/m9.figshare.24556462.v1.

**Funding:** This work was financially supported by a Discovery Grant from the Natural Sciences and Engineering Research Council of Canada

## Abstract

Human-coyote conflict can arise when coyotes follow, pursue or attack pets or people. Although such attacks are rare, they are typically highly publicized, and leave residents concerned about the presence of coyotes in their neighborhoods. Wildlife managers often promote hazing to mitigate human-coyote conflicts, but this technique for intimidating animals has been studied in coyotes only recently and there are few guidelines for its implementation. We developed a community-based hazing program in Edmonton (Alberta, Canada) implemented by volunteers who patrolled their neighborhoods while recording coyotes and coyote attractants. When coyotes were observed, volunteers walked towards the coyotes and recorded their overt reaction and flight initiation distances. If coyotes did not retreat when volunteers were within 40 m of the animal, volunteers ran towards the coyote while shouting and throwing weighted tennis balls towards the animals. Over two field seasons, we recruited, trained, and engaged 120 volunteers from 71 neighborhoods, who conducted 1598 patrols, observed coyotes in 175 instances, and conducted hazing 23 times. Coyotes retreated before volunteers were within 40 m during 124 (71%) of the observations and retreated immediately from 22 (96%) of the hazing events. We found no evidence that hazing changed subsequent measures of overt reaction or flight initiation distances by coyotes, perhaps because it was implemented too rarely, and its effects on the number or timing of subsequent coyote reports by members of the public were inconsistent. Our study emphasizes the rarity of close encounters with coyotes and the high frequency with which they retreat from human approaches and even directed attention by people. Our study supports the use of community-based hazing to reassure members of the public while potentially promoting wariness in coyotes.

## Introduction

Urban coyotes are increasingly common across North America [1] and interactions with them have been described in many communities [2–5]. Coyotes may approach, pursue and attack pets and people [3,6], and den under porches and decks [4], potentially threatening the safety of people and their pets. Although coyote attacks on people remain rare, especially considering the rates of human-coyote interactions within cities [3,4,7], rates of coyote attacks appear

(RGPIN-2017-05915) awarded to CCSC and a Faculty of Science Fellowship to CCSC, and a Biodiversity Grant from the Alberta Conservation Association (0300090140) to GL. The funders did not play any role in the study design, data collection and analysis, decision to publish, or preparation of the manuscript. Natural Sciences and Engineering Research Council of Canada: https://www.nserc-crsng.gc.ca/professors-professeurs/grants-subs/dgigp-psigp_eng.asp Alberta Conservation Association: https://www.ab-conservation.com/grants-program/grants-in-biodiversity/overview/.

**Competing interests:** The authors have declared that no competing interests exist.

to be increasing, possibly due to the prevalence of food-conditioning caused by deliberate or accidental feeding (i.e., mismanagement of food attractants such as compost, bird seeds, garbage, roaming pets and others [8]). Recent attacks on humans have occurred in the Canadian cities of Burlington (ON [9]), Calgary (AB [10]), Edmonton (AB [11]) and Vancouver (BC [12]), leaving many people concerned about the presence of coyotes in their neighborhoods [13,14]. Such concerns may be exacerbated by the way these animals are portrayed in print and social media [15–18].

Management plans for urban coyotes frequently recommend the use of aversive conditioning and hazing as humane ways to manage bold, urban coyotes [19]. Aversive conditioning extends the concept of hazing, which is defined as the act of using deterrents to immediately change the behavior of an animal in a conflict situation [20]. Aversive conditioning refers to the repeated and consistent use of deterrents to reduce the occurrence of similar behaviors in similar contexts over longer time periods by teaching animals greater wariness towards people [21,22]. These techniques likely have ancient origins in wildlife management and both dogs and shouting (i.e., deterrents) have been used by Indigenous people in Northern Canada to deter bears from conflict situations [23]. Although aversive conditioning has been applied extensively to elk (*Cervus canadensis* [24]) and black bears (*Ursus americanus* [25]) it has been applied to bold urban coyotes only recently [19,20,26–28] and there are few guidelines for implementing this technique.

Repeated applications of aversive stimuli to wildlife that foster learning through aversive conditioning are typically conducted by trained wildlife professionals in high intensity programs that may use projectiles, loud noises, and pursuit by humans or trained dogs [24,25,29,30]. Such a program was applied to coyotes in the City of Calgary (AB) with some evidence that it trained coyotes to retreat from people [19], but these approaches are often resource intensive [24,31], and the use of high intensity stimuli can be contentious [32,33]. Furthermore, these high intensity treatments are often applied reactively to conflict individuals rather than proactively on coyotes that are not yet highly aggressive. Use of proactive approaches is typically limited by available personnel, but this likely also limits the efficacy of these treatments [19,20]. One potential solution would be to combine these high intensity treatments with lower intensity hazing programs conducted by community members.

To date, only two studies have addressed the use of community-based hazing to manage bold urban coyotes, both in the Denver Metropolitan Area [20,26]. In one, authors developed a program in which trained volunteers and members of the public were encouraged to haze coyotes when appropriate (determined based on coyote behavior and location) by standing tall, shouting and sometimes throwing objects in the direction of the animal [26]. Following a hazing treatment, coyotes most commonly (81/96, 84%) retreated, although average responses were diminished by the presence of a dog. A second program encouraged members of the public to haze coyotes in two public parks via posted educational signs [20]. The authors found no difference in the overlap in time of activity between people and coyotes in treatment and control parks, suggesting that these hazing applications were ineffective at modifying coyote behavior when applied reactively on targeted bold individuals. Both studies concluded that coyotes retreated from hazing in the short-term [26], but this technique did not seem to have long-lasting effects on coyote behavior [20,26].

Community-based hazing offers several advantages for managing bold urban coyotes, including opportunities to: inform community members about the scientific process of aversive conditioning, increase knowledge about the local environment and the places that attract coyotes, foster higher public support for conservation and management of resident wildlife [34], and realize substantial cost saving over use of wildlife professionals. Moreover,

community members can often cover larger areas more often than contracted wildlife professionals and their larger numbers can prevent animal recognition. When treatment is contingent on particular individuals that animals recognize and avoid, conflict behaviors are likely to continue in the presence of other people [34,35]. However, more evidence is needed to understand whether, and how, coyotes generalize their hazing experiences and to quantify coyote reactions in ways that could translate to management goals or metrics of human safety. Such metrics have been applied to other species, as tolerance of and habituation to people by measuring the distance at which animals react to or flee from people. In grizzly bears (*Ursus arctos*), researchers have recorded overt reaction distance as the distance at which an animal exhibits a visual response to an approaching person [36]. In elk, researchers have measured flight response distance from animal retreats [24], which has been generalized as flight initiation distance for dozens of species [37–39].

Here we describe a community-based hazing program called the Urban Coyote Intervention Program (UCIP) that was implemented in Edmonton, AB in 2021 and 2022. We sought to develop and refine the use of hazing by community members from discrete neighborhoods as a tractable and cost-effective tool to reduce conflict by increasing the wariness of coyotes while empowering people to address bold behavior by coyotes in their own neighborhoods. We targeted residential neighborhoods because coyotes induce more concern in this context [40,41], and conducted the program during the coyote breeding season to reduce their subsequent use of residential areas for denning, which is a season and context with high past rates of human-coyote intetractions [42]. Volunteers in treatment neighborhoods were instructed to haze coyotes only within residential neighborhoods, during the day, and for individuals that could be approached to within 40 m (similar to the distance targeted for elk retreat in national parks to maintain the safety of people [24]). We assessed the effects of hazing on coyotes by measuring changes in their overt reaction and flight initiation distances over time and by the frequency and timing of reports made by residents of each neighborhood to the Edmonton Urban Coyote Project website [43] or the City of Edmonton's 311 report database [44]. If associative learning by coyotes occurred in response to persistent use of hazing by people, we predicted that our hazing procedure would cause (a) an increase in the reaction distances by coyotes and (b) a decline in the reports of human-coyote conflict in the treated neighborhoods.

## Materials and methods

### Study area

We studied the responses of coyotes to a community-based hazing program conducted in the City of Edmonton, central Alberta, Canada (53°34'N, 113°25'W). Edmonton is situated in a transition zone between prairie grasslands and the boreal forest [45], has an elevation of 671 m, mean temperatures ranging from -12° C in the winter to 16° C in the summer, and annual precipitation of 446 mm [46]. The human population of Edmonton was approximately 1,010,899 in 2021 over its 685 square kilometers, making it the fifth most populated city in Canada [46,47]. Half (52%) of the city's total area is occupied by residential lands, with over 242 residential neighborhoods within the City of Edmonton [48]. Natural areas comprise about 7% of the city's total area [49]. The city is bisected by the North Saskatchewan River, which is connected to many ravines that combine to constitute the largest contiguous area of municipally-owned urban parkland in North America. This area provides natural habitat for many wildlife species, including white-tailed jackrabbits (*Lepus townsendii*), white-tailed deer (*Odocoileus virginianus*), beavers (*Castor canadensis*), porcupines (*Erethizon dorsatum*), and coyotes (*Canis latrans* [45]).

## Volunteer training and hazing

Volunteers who lived in or were associated with the participating neighborhoods conducted the hazing. We used newsletters, and social media posts to recruit volunteers living in the 43 communities that had the highest rates of reported coyote observations to the Edmonton Urban Coyote Project (EUCP) website or the civic 311 call center between 2018 and 2020. We also described the program on the EUCP website, in webinars given for other purposes, in media interviews, and casual conversations, to raise awareness of this program. Volunteers were trained via a website that provided information about coyote behavior, the goals of the program, features that attract coyotes as food or shelter, hazing techniques, and volunteer safety (Urban Coyote Intervention Program, [50]). The website also introduced volunteers to the data collection forms to guide their coyote patrols, observations of attractants, and hazing events. Before beginning their patrols, all volunteers had to obtain a perfect score on a quiz that assessed knowledge of program goals, measures of success, key concepts, and the hazing techniques.

Volunteers participated in the hazing program by patrolling their neighborhoods and responding to reports of coyotes by other people. Volunteers developed their own patrol schedules based on their availability, but were encouraged to conduct one or more patrols per week. While patrolling, volunteers noted the time at which they started and ended their patrols and whether or not they found coyotes or attractants. When one or more coyotes was observed, volunteers recorded the time, date, location, context (e.g., number of coyotes, presence of vulnerable individuals, such as children under 12 years old or pets) and behavior of animals (e.g., moving or stationary). To assess coyote wariness, volunteers estimated overt reaction distance (i.e., the distance at which a coyote visibly responds to the presence of a person with a change in position, movement, or attention relative to the position of the observer) and flight initiation distance (i.e., the distance at which a coyote starts to flee when approached) as they began walking slowly towards the coyote while maintaining a fixed gaze on the animal. Volunteers could estimate distances approximating 20 meters or less knowing that 10 meters is equivalent to approximately 14 walking steps [50]. Other distances were estimated using standardized cut-out rectangles on a cards that volunteers held at arm's length; volunteers noted the rectangle that best surrounded the coyote they observed, and recorded the associated distance (20 meters, 40 meters, and 60 meters). If the coyote did not retreat when volunteers were within 40 m of the animal, volunteers were instructed to haze the coyote. A hazing event consisted of running towards the coyote while shouting and throwing modified tennis balls in the direction of the animal. Tennis balls were modified by adding sand to make them the weight of baseballs to increase throwing accuracy, and fitting them with three streamers of pink flagging tape to increase their animation and resemble fladry [51,52]. Volunteers recorded the direction (i.e., as an angle relative to their own approach) and the behavioral response of coyotes (i.e., turn and gallop away, trot away, face the volunteer while backing away, remain in place, or approach, [50]). As part of their patrols or hazing events, volunteers recorded potential coyote attractants, such as accessible compost or garbage, spilled bird seed, and piles of wood that might shelter rodents. Volunteers were encouraged to record attractants during patrols even if a coyote was not observed [50].

The program occurred during two field seasons from January to May 2021 and 2022. In 2021, neighborhoods were assigned as treatment or control; to the extent possible, neighborhoods were paired based on spatial proximity to one another and similar proximity to the river valley and ravine parkland. Hazing was only conducted in treatment neighborhoods. Volunteers in control neighborhoods recorded coyote presence and responses, but did not haze coyotes. Instead, when volunteers in control neighborhoods were within 40

m of coyotes, they stopped and retreated. Volunteers in treatment neighborhoods applied hazing as described above. Because there were few opportunities to conduct hazing in 2021, we eliminated control neighborhoods in 2022 to increase observations of coyote responses to conditioning.

Public reports of coyote sightings and encounters in the City of Edmonton were collected from the EUCP website between January 1, 2021 and June 1, 2022; we also collected public reports of coyotes from the City of Edmonton 311 civic database via private communication during this period. We identified the neighborhood associated with every report, and eliminated reports for which no date was provided. A Research Permit and Collection License allowing patrolling by volunteers on public land and the use of hazing when appropriate was obtained from the Government of Alberta in 2021 and 2022. This study was conducted under the approval of the University of Alberta Animal Care and Use Committees (AUP00003783).

## Data analysis

To assess how the program was implemented by volunteers, we summarized the number of volunteers, duration of coyote patrols, and number of coyotes observed in each neighborhood. We used unpaired Mann-Whitney U tests to determine whether there were differences in the average duration of patrols and the number of coyotes volunteers observed between treatment and control neighborhoods in 2021. We employed Pearson's chi-squared test to determine whether certain behavioral responses (i.e., remained in place, backed away, walked away, trotted away or ran away) occurred more or less frequently than expected depending on the hazing actions conducted by volunteers.

We used logistic regression mixed models [53] to determine whether hazing increased the subsequent overt reaction distances expressed by coyotes. We first grouped overt reaction distances in binary categories of 0-39 m (the distances that were to elicit hazing; assigned 0) or > 40 m (exceeding the distance for hazing; assigned 1). As predictors (i.e., fixed effects) for this binary response variable, we evaluated in four separate models the number of times (within the previous 30 days and within each neighborhood) the overt reaction distances were evaluated, hazing was conducted, attractants were reported, and the number of coyotes present while the overt reaction distance was being evaluated by volunteers; neighborhood was added to each model as a random effect. We used the same approach to determine the effect on flight initiation distances (again classified as less than (0) vs. equal to or more than (1) 40 m) of the same four predictor variables; previous measures of hazing, overt reaction distance, attractants, and number of coyotes present, with each of the four models including neighborhood as a random effect. For each model, we reported the beta coefficient ($\beta$) and confidence intervals (CI) to emphasize effect size [54]. We evaluated model fit using Nakagawa and Schielzeth's $R^2$, which is adapted to generalized linear mixed-effects models [55]. Finally, we compared our models to a null model using likelihood ratio tests. If learning from hazing increased wariness via measures of overt reaction or flight initiation distances, we expected positive coefficients for the number of previous measures of overt reaction distance, previous hazing, and the number of coyotes present. If wariness learned from hazing is opposed by food conditioning, we expected reports of attractants to reduce measures of overt reaction and flight initiation distances.

We investigated the effects of time of day on overt reaction distances using Pearson's chi-squared test. We classified overt reaction distances as less than vs. equal to or more than 40 m. We coded the time of coyote observations as morning (4am - <10am), day (10am - <4pm), evening (4pm- <10pm), or night (10pm - <4am [56]).

We assessed the effects of hazing on coyote reports using the frequency and timing of reports made to the EUCP website and 311 civic call center. To assess frequency, we counted the number of reports made in each neighborhood for the two weeks before and after the use of hazing. We compared these reports to reports made (a) in neighborhoods that participated in the program and where coyotes were observed by volunteers, but hazing was not conducted (i.e., designated control neighborhoods in 2021, and neighborhoods that were not designated as controls, but where hazing was not conducted in 2022) or (b) in neighborhoods that did not participate in the intervention program, but in which reports of coyote activity were submitted. Neighborhoods were paired based on spatial proximity to one another and similar proximity to the river valley and ravine parkland. For neighborhoods without hazing events, we counted coyote reports before and after the date for hazing in the treated neighborhoods. We used negative binomial mixed models to determine whether the number of coyote reports made to either the EUCP or the 311 civic call center significantly differed between the two weeks preceding and the two weeks following each event in neighborhoods where hazing was conducted compared to neighborhoods where it was not. As potential predictors, we explored the effects of time period (i.e., before or after the event) and treatment type (i.e., hazing or control) on the number of reports, as well as the interaction between time period and treatment type, with neighborhood pairs as a random effect. All fixed effects were coded as factors, with "before" and "control" as the reference categories. For each model, we reported the $\beta$, CIs, and Nakagawa and Schielzeth's $R^2$. We compared our models to null models using likelihood ratio tests.

To assess timing, we counted the days to the next report after a hazing event in a neighborhood and assigned dates for neighborhoods without hazing events as above. We excluded neighborhood pairs that were not followed by a coyote report from our analysis. We used negative binomial mixed models to determine whether the number of days to the next report differed significantly between neighborhoods where hazing was conducted compared to neighborhoods where it was not. We considered the effect of treatment (i.e., hazing or control) on the number of days to the next report, with neighborhood pairs as a random effect. The treatment type was coded as a factor, with "control" as the reference category. We reported the beta coefficient ($\beta$), confidence intervals (CI), Nakagawa and Schielzeth's $R^2$, and compared models to null models using likelihood ratio tests.

We assessed the difference in timing of patrols conducted by volunteers and coyote activity by summarizing the number of patrols (n = 1,598), coyote observations (n = 175), and coyote reports (n = 190) for each hour of the day. We collected reports of coyote activity made to the Edmonton Urban Coyote Project website between January and May 2021 and 2022 in neighborhoods that participated in the program (i.e., where patrols were conducted by volunteers). We eliminated reports for which no time was provided, as well as reports with imprecise time (e.g., morning, afternoon, night). All statistical analyses were performed in R version 4.1.0 [57].

## Results

In 2021, 59 volunteers in 28 neighborhoods participated in the program by conducting 657 patrols that summed to 571 hours and 37 minutes. The average duration of patrols was about 13% longer in treatment (n = 349 patrols, $\bar{x}$ = 55.2 minutes, SD = 25.7 minutes) than control neighborhoods (n = 308 patrols, $\bar{x}$ = 48.9 minutes, SD = 25.9 minutes; W = 44329, p < 0.001). In 2022, 77 volunteers (16 of them returning from 2021) in 59 neighborhoods participated in the program by conducting 941 patrols that summed to 737 hours and 19 minutes. Over the two years, the total number of volunteers was 120 in 71 neighborhoods in which 1,598 patrols summed to 1,308.9 hours (Fig 1).

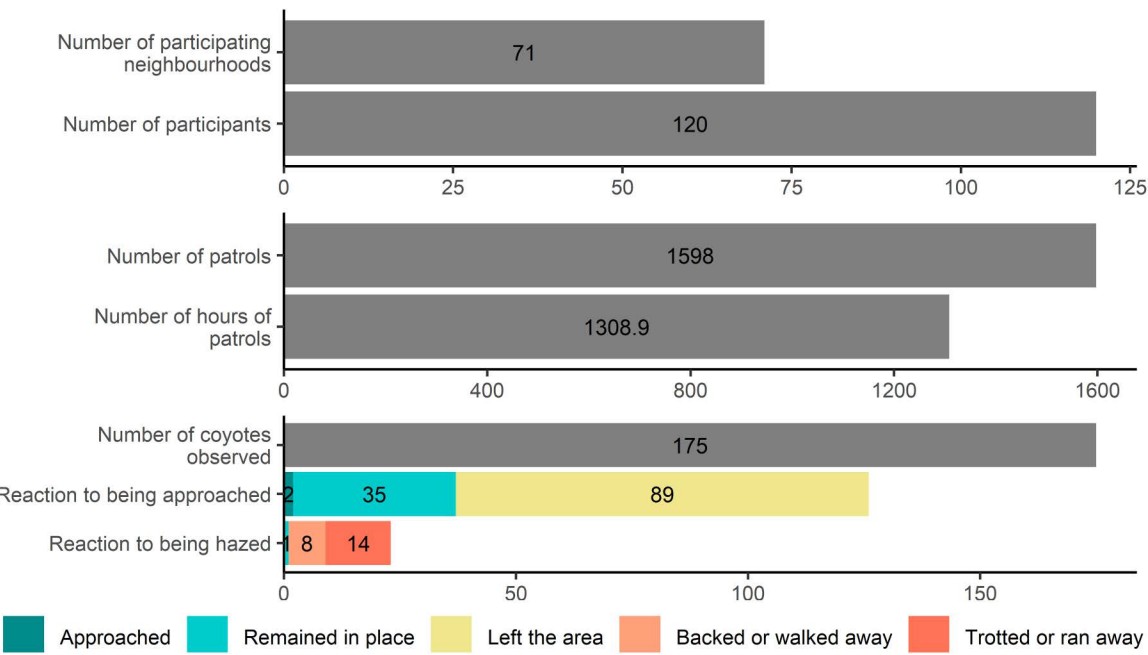

**Fig 1. Volunteer effort in a community-based hazing program conducted in Edmonton, Alberta, in 2021 and 2022 and reactions of coyotes after being approached and hazed by volunteers.**

In 2021, volunteers observed coyotes on 64 occasions in 15 different neighborhoods with almost half (n = 28, 43.8%) occurring in a single control neighborhood (Table S1). An additional 20 observations (31.2%) were made in four other neighborhoods. The mean number of coyote sightings per neighborhood was over six times higher in control (n = 6 neighborhoods, $\bar{x}$ = 8.5 coyotes, SD = 10.0 coyotes) than in treatment (n = 9 neighborhoods, $\bar{x}$ = 1.4 coyotes, SD = 0.5 coyotes) neighborhoods (W = 45.5, p = 0.025). In 2022, volunteers recorded a total of 111 coyote observations in 28 different neighborhoods, again with most concentrated in a few neighborhoods (Table S1).

Across both field seasons, volunteers were able to measure the overt reaction distance in 132 out of the 175 instances (75.4%) where coyotes were observed. Overt reaction distances were more often observed at larger distances ($\bar{x}$ = 42.5 m, SD = 20.4 m), with 22.7% over 60 m (n = 30), 34.8% at 40-60 m (n = 46), 23.5% at 20-39 m (n = 31), 18.2% at 5-19 m (n = 24), and 0.8% less than 5 m (n = 1). When the reaction of coyotes after volunteers measured the overt reaction distance was known (n = 126), over two-thirds of coyotes left the area (n = 89 or 70.6%), but many remained in place (n = 35, 27.8%) and coyotes approached a volunteer or another person on two occasions (1.6%; Fig 1). The flight initiation distance (FID) was measured on 80 occasions. Flight initiation distances were more often observed at intermediate distances ($\bar{x}$ = 38.4 m, SD = 20.2 m), with 18.8% over 60 m (n = 15), 25.0% at 40-60 m (n = 20), 35.0% at 20-39 m (n = 28), 20.0% at 5-19 m (n = 16), and 1.3% less than 5 m (n = 1).

Hazing was conducted 23 times; 5 in 2021 in 4 neighborhoods and 18 times in 2022 in 11 neighborhoods. Hazing actions included shouting (n = 18, 78.3%), running towards the coyote (n = 11, 47.8%), and throwing weighted tennis balls in the direction of or directly at the coyote (n = 6, 26.1%). A volunteer conducted hazing with their leashed dog on one occasion by shouting and running towards the coyote. Following hazing, almost all (22/23, 95.7%) coyotes moved away from the volunteer. On almost half of these occasions, coyotes ran away

from volunteers (10/23, 43.5%), but some walked away (n = 7, 30.4%), trotted away (n = 4, 17.4%), or backed away (n = 1, 4.3%). One coyote remained in place following hazing (Fig 1). There was no association between the hazing treatment used by volunteers and the response of coyotes to that treatment ($X^2$ = 12.65, *P* = 0.24; Table 1).

Volunteers recorded attractants on 695 out of the 1,598 patrols (43.5%) that were conducted across both field seasons. Unsecured garbage was recorded in almost half (n = 293, 42.2%) of the patrols where attractants were observed. Other commonly reported food attractants included prey (i.e., white-tailed jackrabbits, cricetid (Cricetidae) rodents, domestic cats (*Felis cactus*), and American red squirrels (*Tamiascus hudsonicus*, n = 156, 22.4%)), accessible compost (n = 153, 22.0%), fallen fruits (n = 106, 15.3%), and birdseed (n = 96, 13.8%). Volunteers recorded pet food on only 1.6% (n = 11) of the patrols where attractants were reported. Large bushes or low hanging branches were the most commonly recorded shelter attractant (n = 289, 41.6%), followed by accessible sheds, outbuildings or decks (n = 109, 15.7%), piles of trimmed branches or stacked wood (n = 62, 8.9%), and piles of vegetation-based compost, composed of leaves or branches (n = 51, 7.3%).

Neither of the measures associated with coyote wariness, overt reaction and flight initiation distances ≥ 40 m, were influenced by any of the four explanatory variables we considered; the number of times (within the previous 30 days and within each neighborhood) the overt reaction distances were evaluated, hazing was conducted, attractants were reported, and the number of coyotes present while the overt reaction distance was being evaluated by volunteers (Fig 2; Table 2). None of the 8 models performed significantly better than the null models and the comparison between marginal and conditional $R^2$ values revealed that most variation occurred among neighborhoods (Table 2). Overt reaction distances and flight initiation distances also did not differ among times of day (ORD: $X^2$ = 3.24, *P* = 0.36, FID: $X^2$ = 6.47, *P* = 0.09; Table 3).

Our negative binomial mixed regressions revealed a repeated effect of treatment on the number of public reports made to both the EUCP and 311 databases, but the direction of this effect differed by the type of control used (Fig 3). In the neighborhoods with hazing, reports were an average of 47.6% less frequent in the EUCP database (Fig 3A) and 29.3% less frequent in the 311 database than neighborhoods where patrols were conducted, but no hazing occurred (Fig 3B; Table 4). However, when compared to non-participating neighborhoods, reports in neighborhoods with hazing were an average of 3 times more prevalent in the EUCP database (Fig 3A) and 5 times more prevalent in the 311 database (Fig 3B). Beta coefficients for treatment overlapped zero 2 out of 4 times, while the coefficients for time (before vs. after) and the interaction between time and treatment overlapped broadly (Table 4). Three out of four models performed significantly better than the null models, but all models also exhibited much higher values for conditional than marginal $R^2$ values, indicating that most of the variation in these results resulted from variation among neighborhood pairs, relative to the effect of hazing (Table 4).

**Table 1. Hazing actions (n = 23) performed by community scientists between January and May in 2021 and 2022 in Edmonton, Alberta and resulting response of coyotes.**

| Hazing actions performed | Response of coyotes to hazing | | |
|---|---|---|---|
| | Remained in place | Backed or walked away | Trotted or ran away |
| Shouted | 1 | 0 | 2 |
| Ran towards the coyote | 0 | 0 | 1 |
| Threw weighted tennis ball in the direction of the coyote | 0 | 0 | 1 |
| Other | 0 | 2 | 0 |
| Two actions performed | 0 | 4 | 7 |
| Three or more actions performed | 0 | 2 | 3 |

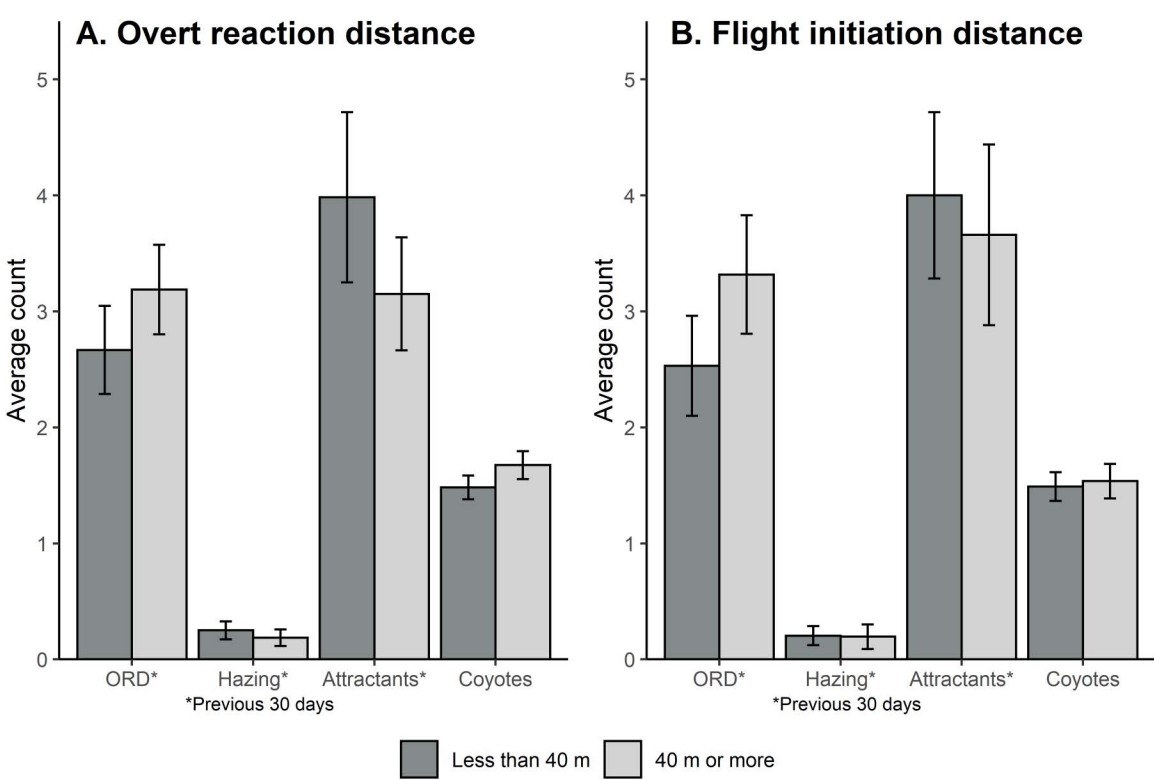

**Fig 2. Average counts** ( ± SE) of the number of times in the previous 30 days the overt reaction distance was measured (ORD), hazing was conducted (hazing), or attractants were recorded in the neighborhood (attractants) for coyotes that were estimated to be less than 40 m vs equal to or greater than 40 m away from observers when they measured overt reaction distance (A) or flight initiation distance (B).

**Table 2. Summary metrics of logistic regression models predicting the overt reaction distances (n = 140) and flight initiation distances (n = 90) of coyotes observed in Edmonton, Alberta, between January and May in 2021 and 2022, with distances classified as less than 40 m (0) vs equal or greater than 40 m (1). Predictors included the past number of times (each measured within the previous 30 days and within a neighborhood) the overt reaction distance (overt reaction distance) was measured, hazing was conducted (hazing), or attractants were recorded (attractants), as well as the number of coyotes present during the event being evaluated (number of coyotes). The beta coefficients (β), odds ratios (OR), 95% confidence intervals (CI), Nakagawa and Schielzeth's marginal and conditional R²** are presented for each model. Model performance was compared to a null model using likelihood ratio tests with *P* < 0.05 as a significance level, and we reported whether there was a difference from the null (Y) or not (N). The table excludes the random effect of neighborhood.

| Predictor | β | CI (β) | P | Marginal R² | Conditional R² | OR | CI (OR) | Vs. null |
|---|---|---|---|---|---|---|---|---|
| Overt reaction distance | | | | | | | | |
| Overt reaction distance | 0.02 | -0.11, 0.16 | 0.74 | 0.001 | 0.049 | 1.02 | 0.90, 1.17 | N |
| Hazing | -0.25 | -0.81, 0.30 | 0.38 | 0.007 | 0.072 | 0.78 | 0.45, 1.36 | N |
| Attractants | -0.04 | -0.12, 0.03 | 0.20 | 0.016 | 0.086 | 0.95 | 0.89, 1.02 | N |
| Number of coyotes | 0.18 | -0.21, 0.56 | 0.37 | 0.008 | 0.058 | 1.19 | 0.81, 1.75 | N |
| Flight initiation distance | | | | | | | | |
| Overt reaction distance | 0.08 | -0.05, 0.21 | 0.24 | 0.019 | NA | 1.08 | 0.95, 1.24 | N |
| Hazing | -0.02 | -0.70, 0.65 | 0.95 | 0.000 | NA | 0.98 | 0.50, 1.91 | N |
| Attractants | -0.01 | -0.10, 0.07 | 0.75 | 0.001 | NA | 0.99 | 0.91, 1.07 | N |
| Number of coyotes | 0.06 | -0.40, 0.52 | 0.81 | 0.001 | NA | 1.06 | 0.67, 1.68 | N |

**Table 3. Distribution of overt reaction distances (*n* = 140) and flight initiation distances (n = 90) presented by coyotes as evaluated by volunteers in Edmonton, Alberta, between January and May in 2021 and 2022 by time of day. Time of day was classified as morning (4am - <10am), day (10am - <4pm), evening (4pm-<10pm), or night (10pm - <4am).**

| | Overt reaction distances | | Flight initiation distances | |
|---|---|---|---|---|
| Time of day | > 40 m | 40 m | > 40 m | 40 m |
| Morning | 19 | 35 | 13 | 21 |
| Day | 23 | 26 | 19 | 12 |
| Evening | 17 | 19 | 16 | 8 |
| Night | 1 | 0 | 1 | 0 |

The number of days to the next report also produced conflicting results, this time with similarity between the two control types, but differences in direction between the two databases. For the EUCP database, the number of days following a hazing event to the next report was 28.4% higher than in neighborhoods with no hazing or those that did not participate (Fig 4). For the 311 database, the number of days until the next report was 47.6% lower than the no hazing and non-participating neighborhoods (Fig 4). Statistically, only one of our two models performed better than the null model, again the much higher values for conditional than marginal $R^2$ indicated that most of the variance was attributable to neighborhood pairs (Table 5). When examining the two control types separately, we found that the number of days following a

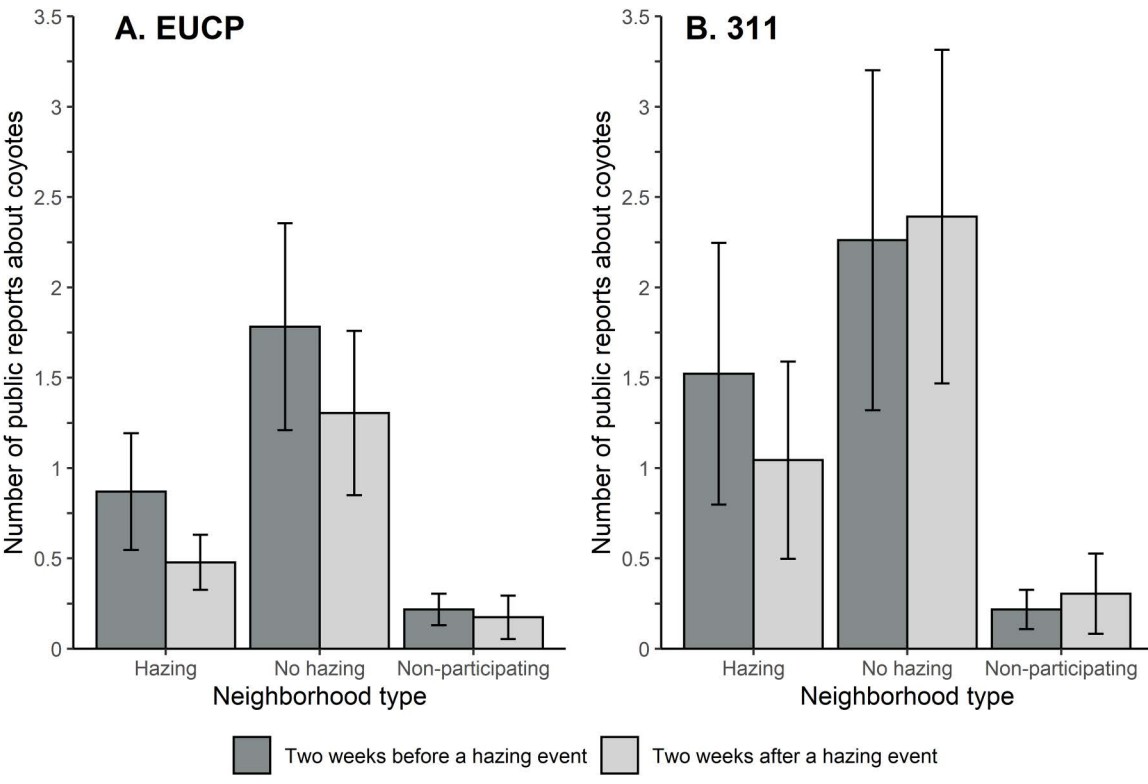

**Fig 3. Average number of coyote reports ( ± SE) made to the Edmonton Urban Coyote Project (EUCP) website (A) or 311 civic call center (B) in two-week periods before and after each hazing event.** Coyote reports were compared between three neighborhood types; where hazing was conducted (hazing), in participating neighborhoods where coyotes were observed by volunteers but hazing was not conducted (no hazing), and in non-participating neighborhoods (non-participating).

**Table 4. Summary metrics of negative binomial mixed regression models predicting the number of coyote reports made in each neighborhood to the Edmonton Urban Coyote Project website (EUCP reports) or to the 311 civic call center (311 report) for the two weeks before and after the use of hazing. Counts of reports encompassed the same dates in participating neighborhoods where coyotes were observed but hazing was not conducted (control: no hazing conducted) and in non-participating neighborhoods (control: non-participating neighborhoods). Potential predictors included the time period considered (i.e., before or after) and the event type (i.e., control or hazing), classified as factors with "before" and "control" as the reference categories. The beta coefficients (β), rate ratios (RR), 95% confidence intervals (CI), Nakagawa and Schielzeth's marginal and conditional R² are presented for each model. Model performance was compared to a null model using likelihood ratio tests with P < 0.05 as a significance level, and we reported whether models were significantly different from the null (Y) or not (N). The table excludes the random effect of neighborhood pairs.**

| Predictor | β | CI (β) | P | Marginal R² | Conditional R² | RR | CI (RR) | Vs. null |
|---|---|---|---|---|---|---|---|---|
| EUCP reports, control: no hazing conducted | | | | | | | | |
| After | -0.33 | -1.06, 0.39 | 0.37 | 0.065 | 0.461 | 0.72 | 0.34, 1.47 | Y |
| Treatment | -0.65 | -1.41, 0.11 | 0.10 | | | 0.52 | 0.24, 1.12 | |
| After * Treatment | -0.26 | -1.42, 0.91 | 0.67 | | | 0.77 | 0.24, 2.48 | |
| EUCP reports, control: non-participating neighborhoods | | | | | | | | |
| After | -0.26 | -1.64, 1.12 | 0.71 | 0.104 | 0.350 | 0.77 | 0.19, 3.07 | Y |
| Treatment | 1.30 | 0.24, 2.37 | 0.02 | | | 3.68 | 1.27, 10.65 | |
| After * Treatment | -0.28 | -1.91, 1.34 | 0.73 | | | 0.75 | 0.15, 3.81 | |
| 311 reports, control: no hazing conducted | | | | | | | | |
| After | -0.08 | -1.13, 0.97 | 0.88 | 0.011 | 0.885 | 0.92 | 0.32, 2.63 | N |
| Treatment | -0.35 | -1.41, 0.71 | 0.52 | | | 0.71 | 0.24, 2.05 | |
| After * Treatment | -0.51 | -2.06, 1.03 | 0.51 | | | 0.60 | 0.13, 2.80 | |
| 311 reports, control: non-participating neighborhoods | | | | | | | | |
| After | 0.16 | -1.31, 1.64 | 0.83 | 0.094 | 0.618 | 1.18 | 0.27, 5.20 | Y |
| Treatment | 1.75 | 0.47, 3.03 | 0.008 | | | 5.75 | 1.59, 20.71 | |
| After * Treatment | -0.67 | -2.47, 1.13 | 0.47 | | | 0.51 | 0.08, 3.11 | |

hazing event to the next report was 9.4% lower than the non-participating neighborhoods, but 64.8% higher than the no hazing neighborhoods for the EUCP database. The direction of these conclusions was unchanged by examining the two control types separately for the 311 database.

When comparing the timing of patrols and coyote observations made by volunteers to that of coyote reports made by members of the public, we found that most patrols (n = 1316, 82.4%) were conducted between 7am and 7pm, with a peak in the morning between 7am and 10 am (n = 456, 28.5%), and in the afternoon between 1pm and 5pm (n = 596, 37.3%; Fig 5A). Coyotes were most commonly observed by volunteers in the morning between 8am and 11am (n = 76, 43.4%; Fig 5B). Although coyotes were reported by members of the public through-out the day, coyote reports were more common between 8am and 10am (n = 35, 18.4%), and between 8pm and 11pm (n = 49, 25.8%; Fig 5C).

## Discussion

Although encounters with bold urban coyotes appear to be increasing across the continent [1,4,40], there is widespread public resistance to culling [58,59]; this creates an urgent need to identify behavior-based approaches for reducing negative interactions. We designed and implemented a community-based hazing program to determine whether it could increase coyote wariness in the late winter when coyotes are breeding and potentially establishing territories in residential areas. We recruited, trained, and engaged 120 volunteers from 71 residential neighborhoods, who spent over 1,308 hours in 2021 and

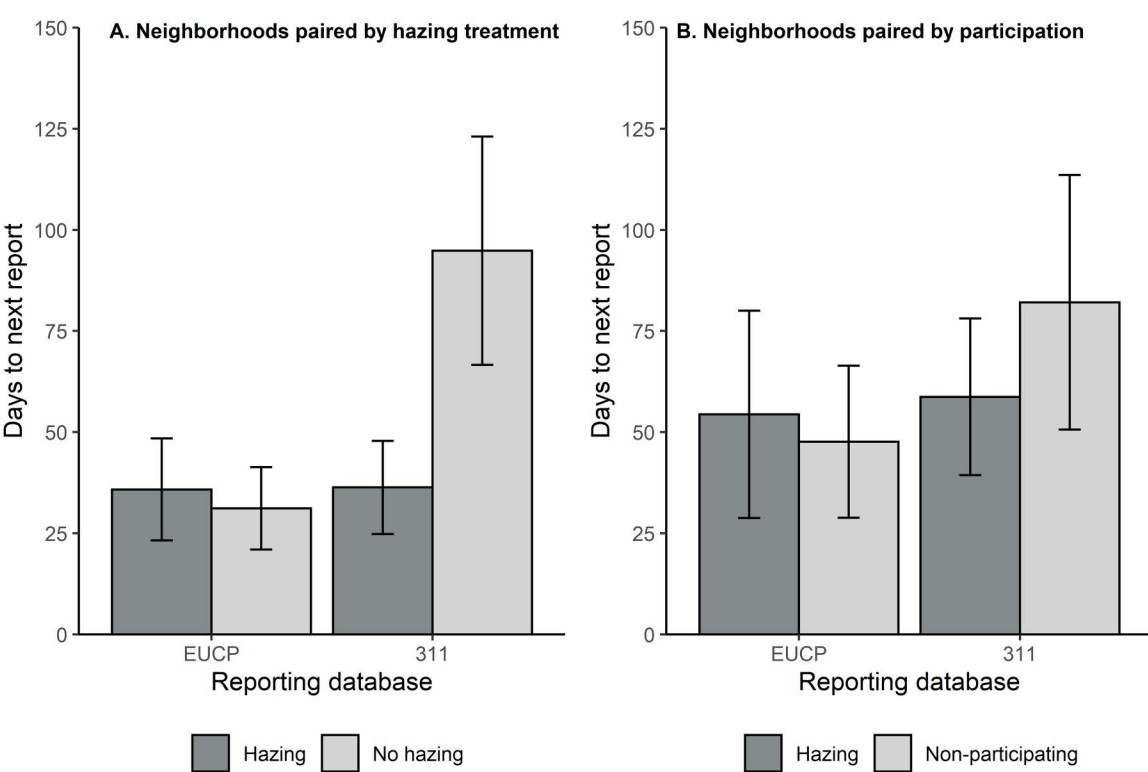

**Fig 4. Average number of days ( ± SE) to the next coyote report made to the Edmonton Urban Coyote Project (EUCP) website or 311 civic call center for neighborhoods with a hazing event compared to paired control neighborhoods with no hazing** (A) or non-participating neighborhoods (B). We excluded neighborhood pairs where no coyotes were reported following the hazing event.

**Table 5. Summary metrics of negative binomial mixed regression models predicting the number of days to the next report made to the Edmonton Urban Coyote Project website (Days until the next EUCP reports) or to the 311 civic call center (Days until the next 311 report) following a hazing event conducted in Edmonton, Alberta, between January and May in 2021 and 2022. Predictor variables compared matched pairs of neighborhoods in which one had a hazing event and the other was a control (reference category), which combined no hazing and a non-participating neighborhoods. Each model includes the beta coefficients (β), rate ratios (RR), 95% confidence intervals (CI), Nakagawa and Schielzeth's marginal and conditional R². Model performance was compared to a null model using likelihood ratio tests with $P < 0.05$ as a significance level to determine whether models were significantly different from the null (Y) or not (N). The table excludes the random effect of neighborhood pairs.**

| Predictor | β | CI (β) | P | Marginal R² | Conditional R² | RR | CI (RR) | Vs. null |
|---|---|---|---|---|---|---|---|---|
| Days until the next EUCP report | | | | | | | | |
| Hazing | 0.25 | -0.24, 0.74 | 0.32 | 0.009 | 0.507 | 1.28 | 0.79, 2.09 | N |
| Days until next 311 report | | | | | | | | |
| Hazing | -0.65 | -1.18, -0.12 | 0.016 | 0.047 | 0.507 | 0.52 | 0.31, 0.89 | Y |

2022 patrolling for coyotes and coyote attractants in their neighborhoods. Volunteers reported attractants on 695 occasions; unsecured garbage and large bushes or low hanging branches were the most commonly recorded attractants. Although volunteers observed coyotes on 175 occasions, it was possible and appropriate to apply hazing only 23 times by shouting, throwing weighted tennis balls, and chasing coyotes. Most coyotes (70.6% of 175 occasions) retreated after volunteers determined the overt reaction distance, which necessarily included a fixed gaze on the coyote, and almost all coyotes (95.7% of 23 events) retreated from hazing. We found little evidence that overt reaction and flight initiation distances changed with the number of previous times in that neighborhood

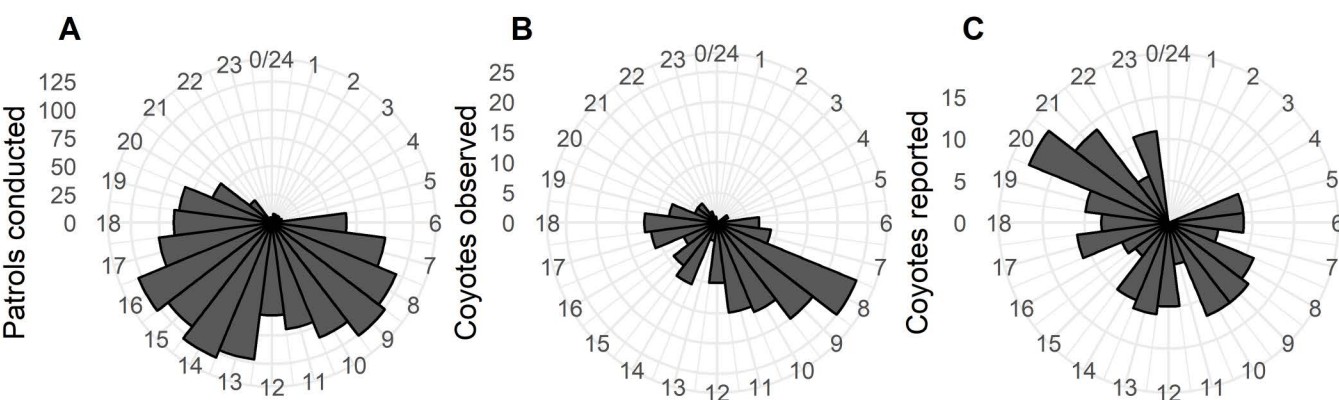

**Fig 5. Rose plots describing timing of patrols** (A), coyote observations (B), and coyote reports (C), binned by hour of the day and multiples of 25, between January and May in 2021 and 2022 in participating neighborhoods. Plots exclude reports for which no time was provided and reports with imprecise times (e.g., morning, afternoon).

that overt reaction distances were measured, hazing was conducted, or attractants were reported. Similarly, there was no consistent effect of treatment on the number of coyote reports; hazing events increased the number of days before the next public report of coyotes in a research database, but reduced those values for reports to a civic call center. Overall, our program revealed that coyotes were seen rarely and readily retreated from people, but we did not find consistent effects of our program on subsequent coyote behavior or public reporting rates.

Perhaps the most important finding from our study is the relative rarity with which people encountered coyotes, even in neighborhoods with high relative rates of previous reporting and where people were actively patrolling for coyotes. Our protocol required that people patrol only during the day and in residential areas because these are the circumstances that have most alarmed people in the past [40,41]. Our finding that only 7.4% of patrols led to a coyote observation provides useful context for understanding the increasing prevalence of media reports that portray bold coyotes in residential areas [16]. When coyotes were observed, more than half (57.6%) presented an overt reaction distance that we considered appropriately shy (over 40 m) and most coyotes (70.6%) retreated after being approached at a walk by volunteers. Several authors have reported that urban coyotes typically exhibit avoidance or indifference towards people [2–4,40]. Our findings suggest that social media posts and conventional media reporting of rare events may lead people to overestimate the prevalence of bold or aggressive behavior by coyotes. Other authors have suggested that quantitative information about the low probability of negative encounters could increase public acceptance of coyotes in cities [15,16,18].

The prevalence of coyote observations that supported the use of hazing in our study was remarkably similar, both 13%, to one of the only two studies of community-based hazing in coyotes, which occurred in Denver, CO [26]. The proportion of coyotes that retreated from hazing is also similar to the Denver study (95.7% in our study, 84.4% for Bonnell & Breck, 2017). Although they received the same instructions, our volunteers varied in their use of hazing treatments consisting of running towards coyotes, shouting, throwing weighted tennis balls in the direction of the animals, or by combining these tools. We found no differences in the responses of coyotes to these treatments, but small sample sizes with multiple treatment categories limited the power of our test. Learning theory suggests that more intensive stimuli are more likely to generate a sensitization response to conditioning and prevent habituation

[30,60]. Our results suggest that community-based hazing is an effective tool for deterring urban coyotes during an encounter, as suggested by others [20,26].

We found no effect of past approaches by people or hazing on two measures of coyote boldness; overt reaction distance and flight initiation distance. We expected both values to increase if coyotes learned to be more wary of people after hazing, consistent with the prediction of a learned association with negative stimuli in association with people [60]. Increases in flight initiation distance following aversive conditioning have been reported in elk [24,29] and bears [61]. Similarly, we did not find evidence that the number of reports of attractants reduced measures of wariness via overt reaction of flight initiation distances, as would be expected if attractants produced food conditioning that made animals more tolerant of people [21,62]. Our definition of attractants was broad and may not have had enough spatial and temporal precision to associate them with coyote behavior. Because food conditioning has been linked to an increased likelihood of attacks on people by coyotes, it remains important to remove or secure attractants in residential areas [1,3,63].

The lack of evidence we found for learned responses to previous approaches or hazing by people or reports of attractants would be expected if coyote density was either very high or very low, such that volunteers rarely encountered and residents rarely reported the same coyotes that had been observed or treated previously. We do not know the density of coyotes in our study neighborhoods, but a previous study of GPS-collared coyotes in Edmonton reported large variation in home range sizes, from 1.4-109.0 km$^2$ [49]. Moreover, coyotes that used residential neighborhoods more often included diseased animals that appeared to range very broadly [64]. In the Chicago Metropolitan Area, the average home range of transient coyotes (mean = 26.80 km$^2$) was five times larger than the home range of resident coyotes (mean = 4.95 km$^2$ [65]). If our volunteers typically encountered transient animals, it is very likely that they did not encounter the same animals on successive measurements and those animals may, similarly, have encountered a wide diversity of attractants to prevent evidence of habituation at the scale of neighborhood reports.

Even if coyotes made repeated use of neighborhoods in our study, hazing may have been too infrequent to generate learning. In our program, only 1.1% of patrols led to a hazing event and hazing was mostly conducted only once or twice in each neighborhood, providing coyotes with very few opportunities to learn from those events and generalize their experience to other contexts. We found an inconsistent effect of hazing on the number of reports by the public; fewer reports occurred in neighborhoods with hazing compared to participating neighborhoods where no hazing was conducted, but more in comparison to non-participating neighborhoods. It is also possible that the fewer reports from neighborhoods with hazing was caused by similar coyote responses to the measurement of overt reaction distances and the very low rates of reporting in non-participating neighborhoods reflected the overall lack of coyote sightings in these areas. Changes in the timing of reports following hazing were inconsistent between the two reporting databases; slightly later in the EUCP database, but earlier in the 311 database. There may be a tendency for people to report bold coyotes to 311 more proactively in the neighborhoods targeted for this study, which exhibited high previous prevalence of reports. Taken together, our results about the effect of hazing on public reporting are similar to community-based hazing programs for urban coyotes in Denver, where no longer-term effects of the treatment were observed [20,26]. In other programs employing aversive conditioning, marked animals make it possible to target individuals repeatedly and more consistently [24,25,29]. Community-based programs will likely be most effective if they are combined with other management tools that produce longer-term changes in coyote behavior, such as high intensity aversive conditioning conducted by wildlife professionals [19], and the lethal management of targeted, problem individuals [20].

Our results should be interpreted with awareness of several limitations. First, our program only operated during daylight hours and was restricted to residential neighborhoods, and excluded natural areas. This approach targeted areas with high past rates of conflict, and maximized the safety of volunteers and coyotes. Other studies suggest that coyotes increase nocturnal behavior to reduce interactions with people [66,67], which occurs in many other carnivores [68,69]. Nonetheless, there may be more opportunities to encounter and treat coyotes at times when people are less active [5,70–72], as suggested by the high frequency of coyote reports between 20:00 and 23:00 when it was dark and no patrols occurred. We also avoided hazing in natural areas, which provide a refuge for many species and where coyotes provide ecological services and aesthetic enjoyment for people [41,73]. However, several other studies have shown that coyote sightings and encounters are more common in open or natural areas [4,40,74,75]. Even within the spatial and temporal contexts of our study, we had an insufficient number of volunteers to make it likely that coyotes occupying residential areas were always detected and treated. Finally, all of our data was collected during the breeding season, which is less associated with human-coyote conflicts than the pup-rearing season [3,7,40,76,77].

## Conclusion

Our study demonstrated several important results that could inform coyote management and future studies. First, we experienced tremendous enthusiasm from our volunteers who represented about one-third (29.3%) of the communities in Edmonton; there was also a high degree of public interest in coyotes and in our hazing program in the media throughout the study period. Second, our study quantified the rarity (3.5% of patrols) with which people encountered coyotes at distances as close as 40 m in residential Edmonton, even while looking for them in neighborhoods with relatively high past rates of reporting. Third, we showed that coyotes almost always retreated from people when they were treated aggressively (i.e., when being hazed) and over two-thirds of the time when people simply walked towards them while looking directly at them. Our results describing the effects of hazing on coyote behavior and public reporting were inconclusive, likely owing to limitations of sample size and spatial and temporal precision, indicating that much more work is needed to determine the responses of coyotes to proactive hazing by people. Particularly important are assessments of low- vs. high-intensity stimuli that might produce greater evidence of longer-term changes in coyote behavior [19]. Learning theory predicts that aversive conditioning is more likely to be effective if it is immediate, intensive, consistent, evolutionarily relevant, and easily predicted in advance [78,79]. Techniques that can be used by dog-walkers should be explored because dogs are frequently involved in human-coyote conflict [15,40,77,80] and because the presence of a dog appears to reduce the likelihood that a coyote retreats from an aversive conditioning event [26,28]. Complementary tools for managing urban coyotes will likely always be necessary. Others have shown that education campaigns can reduce conflict and increase public confidence around coyotes [81,82]. Food attractants promote conflict and must be removed or secured [1,3], and lethal removal of targeted problem individuals is sometimes necessary to maintain public safety [20].

## Supporting information

**Table S1. Summary of all coyote observations made by volunteers of a community-based hazing program in Edmonton, Alberta, between January and May 2021 and 2022.**
(DOCX)

## Acknowledgments

We thank the City of Edmonton for their conceptual support, 71 participating communities and 120 participating volunteers; C. Allen, K. Andrusiak, J. Bogner, D. Bratle-Kendall, D. Brochu, J. Brohman, C. Burt, A. Cain, D. Cartwright, S. Copen, S. Cribbs, D. Currie, J. Der, B. Geiger, C. Gibson, P. Gillanders, W. Hoban, A. Horon, M. Huyb, G. Kent, L. Kraychy, S. Lambert, C. LoCicero, D. Luxton, E. Marshall, D. McConnell, B. Neil, C. Neilson, A. Petty, M. Qureshi, M. Ritchie, L. Romanchuk, I. Roth, V. Sharma, K. Stevens, S. Storvold, E. Thornton, D. Treasure, P. Venegas Garcia, B. Walker, C. Chang-Yen Phillips, and others who preferred to remain anonymous. We thank the Saville Tennis Centre for providing used tennis balls for volunteers and S. Clarkson for helping to make them into coyote deterring devices.

## Author contributions

**Conceptualization:** Gabrielle Lajeunesse, Howard W. Harshaw, Colleen Cassady St. Clair.

**Data curation:** Gabrielle Lajeunesse.

**Formal analysis:** Gabrielle Lajeunesse, Colleen Cassady St. Clair.

**Funding acquisition:** Gabrielle Lajeunesse, Colleen Cassady St. Clair.

**Investigation:** Gabrielle Lajeunesse, Howard W. Harshaw, Colleen Cassady St. Clair.

**Methodology:** Gabrielle Lajeunesse, Howard W. Harshaw, Colleen Cassady St. Clair.

**Project administration:** Gabrielle Lajeunesse, Howard W. Harshaw, Colleen Cassady St. Clair.

**Resources:** Howard W. Harshaw, Colleen Cassady St. Clair.

**Supervision:** Howard W. Harshaw, Colleen Cassady St. Clair.

**Validation:** Howard W. Harshaw, Colleen Cassady St. Clair.

**Visualization:** Howard W. Harshaw, Colleen Cassady St. Clair.

**Writing – original draft:** Gabrielle Lajeunesse, Howard W. Harshaw, Colleen Cassady St. Clair.

**Writing – review & editing:** Gabrielle Lajeunesse, Howard W. Harshaw, Colleen Cassady St. Clair.

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
