## [Decision Letter · Decision Letter 0]

27 Sep 2024

PONE-D-24-06896An urban coyote intervention program reveals bold coyotes to be observed rarely and retreat consistently from approaches by trained volunteers in residential neighborhoods with high previous rates of coyote reportsPLOS ONE

Dear Dr. St. Clair,

Thank you for submitting your manuscript to PLOS ONE. After careful consideration, we feel that it has merit but does not fully meet PLOS ONE’s publication criteria as it currently stands. Therefore, we invite you to submit a revised version of the manuscript that addresses the points raised during the review process.

This manuscript has the potential to provide a valuable contribution to the literature on human-wildlife conflict in residential areas. Two experts in the field have provided helpful comments for your revisions. The reviewers' main concerns center on the needs for additional information to ensure that the reader can follow the analyses, as well as other details and clarifications needed throughout the text. 

We look forward to receiving your revised manuscript.

Kind regards,

Stephanie S. Romanach, Ph.D.

Academic Editor

PLOS ONE

“We thank the City of Edmonton for their conceptual support, 71 participating communities and 120 participating volunteers; C. Allen, K. Andrusiak, J. Bogner, D. Bratle-Kendall, D. Brochu, J. Brohman, C. Burt, A. Cain, D. Cartwright, S. Copen, S. Cribbs, D. Currie, J. Der, B. Geiger, C. Gibson, P. Gillanders, W. Hoban, A. Horon, M. Huyb, G. Kent, L. Kraychy, S. Lambert, C. LoCicero, D. Luxton, E. Marshall, D. McConnell, B. Neil, C. Neilson, A. Petty, M. Qureshi, M. Ritchie, L. Romanchuk, I. Roth, V. Sharma, K. Stevens, S. Storvold, E. Thornton, D. Treasure, P. Venegas Garcia, B. Walker, C. Chang-Yen Phillips, and others who preferred to remain anonymous. Funding was provided by the Alberta Conservation Association (0300090140) and the Natural Science and Engineering Research Council (RGPIN-2017-05915). This study was conducted under the approval of the University of Alberta Animal Care and Use Committees (AUP00003783).”

“This work was financially supported by a Discovery Grant from the Natural Sciences and Engineering Research Council of Canada (RGPIN-2017-05915) and a Faculty of Science Fellowship to CCSC, and a Biodiversity Grant from the Alberta Conservation Association (0300090140) to GL. The funders did not play any role in the study design, data collection and analysis, decision to publish, or preparation of the manuscript.

Natural Sciences and Engineering Research Council of Canada: https://www.nserc-crsng.gc.ca/professors-professeurs/grants-subs/dgigp-psigp_eng.asp

Alberta Conservation Association: https://www.ab-conservation.com/grants-program/grants-in-biodiversity/overview/”

7. We note that there is identifying data in the Supporting Information file < Table_1_SuppInfo.docx>. Due to the inclusion of these potentially identifying data, we have removed this file from your file inventory. Prior to sharing human research participant data, authors should consult with an ethics committee to ensure data are shared in accordance with participant consent and all applicable local laws.

-Location data

Reviewers' comments:

Reviewer's Responses to Questions

**Comments to the Author**

1. Is the manuscript technically sound, and do the data support the conclusions?

Reviewer #1: Yes

Reviewer #2: Yes

2. Has the statistical analysis been performed appropriately and rigorously? 

Reviewer #1: Yes

Reviewer #2: Yes

3. Have the authors made all data underlying the findings in their manuscript fully available?

Reviewer #1: Yes

Reviewer #2: Yes

4. Is the manuscript presented in an intelligible fashion and written in standard English?

Reviewer #1: Yes

Reviewer #2: Yes

5. Review Comments to the Author

Reviewer #1: Title

The title is a bit too long, I suggest to make it shorter

Introduction

Line 118. I would elaborate more the predictions of your study, you run a lot of analyses which are not expected based on these predictions.

Methods

Line 154. You talk about attractants here, but for me it was confusing about what you meant. I tought about attractants and baits, like those use to attract animals in nature (like special odors or substances). Then I understood in line 169 what you meant. Evaluate the possibility to explain it before and make it clearer for the reader since the beginning. Moreover, I think it is very interesting to know the list of possible attractants you suggested to the citizen, and what they reported (it might go to the supplementary material?).

Line 156. “vulnerable individuals” and “behavior of animals”: these are not described. I think that in general it would be interesting to include in the supplementary material the instructions you gave to the citizens, which I imagine include the description of animals behaviors, the list of possible attractants and so on.

Line 157: while is it clear to me what the flight initiation distance is because is a common measure in many studies (is the distance at which the subject starts to move away from the person approaching), it is less clear what the “overt reaction distance” might be for a coyote: which behavior did they do? Turning the head toward the person, freezing, changing posture…or?

Line 158: Nice idea. However, I am confused with what you reported in the results where for both the overt reaction and flight initiation distance you also included less than 5 and 5-19 m, but you did not mention that here.

Line 167: what is the difference between back away and run or trot away? Maybe also include this fine description in the supplementary (see my previous comment)

Statistics

General comment: for all the models did you check the models’ assumptions? And what did they reveal? You should also specify clearly if you run GLMM or GLM, even if you said you included the random effect.

Line 192: I would specify which behavioural responses.

Line 198: You said you included one fixed effect for each model, which I suppose are: number of times the “overt reaction distances were evaluated”, “hazing was conducted”, “attractants were reported”? and also number of coyotes?. I suggest to rephrase it a bit a make it clearer. Also report the total number of models here. Moreover, I think is not clear which are the predictions behind the use of certain factors. For example, what do you expect from “number of times the attractant were reported”? Did you have specific predictions in mind for each variable you included in the models?

Line 209: I would specify which other variables you included.

Results

Line 313: you talk about 8 models here but is a bit confusing because when you describe the statistics it is difficult for the reader to understand how many models you run. I think you should help the reader in linking the description of the statistics with the results (and also with the predictions you report in the introduction).

Figure 1: here you reported in the graph the number of coyotes observed (126). But if I understood correctly they are not the total number of coyotes observed but just those for whom the reaction is known. Thus, referring to “Number of coyotes observed” in the picture is a bit misleading.

Line 276. You reported here 31 cases 20-39 m, 24 cases of 5-19 m and 1 case of less than 5 m of over reaction distance, but in the methods above you wrote “ If the coyote did not retreat when volunteers were within 40 m of the animal, volunteers were instructed to haze the coyote.” Why volunteers did not haze the coyotes in those cases? Were those control cases?

Figure 3: What is the unit of measurement of the y-axis?

Line 343-345: “however, when compared to non-participating…”: sorry, this sentence is not clear to me, could you explain me this result further?

Table 4: reading the table it seems to me you have included an interaction in your model between the predictors “time period” and “event type”, but I lost this information in the description of the analyses. And what was the structure of the null model? Was identical to the full model but did you remove all factors and the interaction between them? (In case of interaction between the two factors there is no need in the table to report the P values for the single factors that were included in the interaction, since having limited interpretation.)

Discussion.

Line 421.“included a fixed gaze on the coyote”: this should be mentioned in the methods, it strengthen the fact that the approach of the human could be perceived as a threat by the animal.

Line 422: I think you could discuss a little bit more the fact you did not find a significant effect. For example, it might be because there were too few events of hazing (how many coyotes are estimated in those areas), or that in general coyotes are already quite shy and flee from people…as you wrote in the next paragraph. However, your results do not imply that this method would not work in other context or if improved somehow.

Line 457: It is quite difficult to assess whether the absence of a learning association is really due to the fact that they did not learn…in fact you do not have the subjects’ ID. But, I am wondering, do you have any idea of the density of coyotes in the urban areas? Whether they form packs, or are more solitary, or the sizes of their home range? How much likely is that in the same neighborhood there are always the same individuals or rather people encounter always new subjects?

Line 470: “Our results for the number and timing of future public reports are consistent with a lack of learning”. I am not sure about this, I would not say that they did not learn…I think you do not have enough data to conclude this.

Line 476: How do you explain this inconsistency between EUCP database and 311 database?

Line 480: the results of the study done in Denver might support the absence of associative learning or did also suffer from lack of data? In general, do you think your method could be implemented by recruiting more people, investing more time in patrols and so on?

Line 519-522: I personally agree with you, but adding these sentences here, written in this way, makes me think that the data of your study is supporting this, but this is not the case.

Reviewer #2: I’m curious why you chose 40m? Might have been just a reasonable distance to choose which is fine, just wondering if there was something more behind it (and if so, would be worth a mention).

Page 17, line 351-356. Is all of this the figure title? If so, should all be in bold? If not, a new sentence needs to be used after the semi-colon.

It's worth mentioning early in the paper that measuring overt reaction distances involves some pressure on the coyote (direct staring at the coyote). You mention this in the discussion but I think it would be helpful for readers to know at the outset.

It would be useful to discuss the possibility that hazing for more than two weeks might be necessary to impact coyote behavior (so more chances that the same coyotes will be hazed multiple times, for example?). I think you have really important findings here, but we don’t know the full story quite yet and you should mention that. I know you say that hazing happened very rarely in your study and that there were few opportunities for coyotes to learn from those incidences as a result, but I do think this is a point that deserves some more attention. It doesn’t diminish the importance of your findings but it provides important context and potential avenues for future research.

6. PLOS authors have the option to publish the peer review history of their article (what does this mean?). If published, this will include your full peer review and any attached files.

Reviewer #1: No

Reviewer #2: No

---

## [Author Response · Author response to Decision Letter 1]

14 Nov 2024

Manuscript PONE-D-24-06896

Response to Reviewers

Dear Dr. Romanach,

Thank you for giving us the opportunity to submit a revised draft of our manuscript “Urban coyotes were observed rarely and retreated consistently from assertive approaches by volunteers in neighborhoods”. We appreciated the comments and suggestions made by both reviewers, and have responded to individual comments below, in blue. We believe this revised draft addresses the reviewers concerns by providing additional information on the analyses that were performed and clarifying some of our results by emphasizing our main findings and their significance.

---

## [Editor Report · Decision Letter 1]

19 Nov 2024

PONE-D-24-06896R1Revised title: Urban coyotes were observed rarely and retreated consistently from assertive approaches by volunteers in neighborhoodsPLOS ONE

Dear Dr. St. Clair,

Thank you for submitting your manuscript to PLOS ONE. After careful consideration, we feel that it has merit but does not fully meet PLOS ONE’s publication criteria as it currently stands. Therefore, we invite you to submit a revised version of the manuscript that addresses the points raised during the review process.

Your revisions have improved the manuscript, but the Supporting Information files need a fair amount of attention. It would be helpful to provide metadata, but at the least, units of measure and values for each cell. For example, in the EUCP file, what unit is .875 in the Time column? Is it .875 of an hour? Further, this column contains entries such as "12 Noon" and "evening". This file also has errors, e.g., row 9235. In addition, there are blank cells that would benefit from a missing value designation. In the CoyoteYoung column, what is the difference between Unknown and a blank cell? In the 311 file, there are some -1 values in the Incident_Time column causing errors. The UCIP file, for example, has highlighted fields with no explanation for the color coding. Please carefully review and revise your supplementary files. Ideally, your data files should be machine readable for analysis.

We look forward to receiving your revised manuscript.

Kind regards,

Stephanie S. Romanach, Ph.D.

Academic Editor

PLOS ONE

Journal Requirements:

**Additional Editor Comments:**

Your updated title is an improvement. Please remove "Revised title:" from your next submission.

---

## [Author Response · Author response to Decision Letter 2]

5 Jan 2025

We revised the data file as follows and believe we have met all of your requirements.

1. We removed the highlighting for cells that were highlighted in the UCIP dataset.

2. We removed a few columns (i.e., "Season", "NightDay", "Type", "CoyoteYoung") that we weren't using in the MS and for which there were a lot of blank cells in the EUCP dataset.

3. We changed the date and time format where necessary in the EUCP database. We adjusted the R code to reflect these changes

4. We adjusted the date for row 9235 of the EUCP database. The date was in the wrong format (day/month/year instead of month/day/year). This error was only present in one row.

5. We replaced all the "-1" values in the Incident Time column of the 311 database with blank values.

These changes had no impact on the manuscript so we have not submitted an updated MS. Please let me know if you need anything else.

---

## [Editor Report · Decision Letter 2]

12 Jan 2025

Revised title: 

Urban coyotes were observed rarely and retreated consistently from assertive approaches by volunteers in neighborhoods

PONE-D-24-06896R2

Dear Dr. St. Clair,

We’re pleased to inform you that your manuscript has been judged scientifically suitable for publication and will be formally accepted for publication once it meets all outstanding technical requirements.

Kind regards,

Stephanie S. Romanach, Ph.D.

Academic Editor

PLOS ONE
---

## [Editor Report · Acceptance letter]

PONE-D-24-06896R2

PLOS ONE

Dear Dr. St. Clair,

I'm pleased to inform you that your manuscript has been deemed suitable for publication in PLOS ONE. Congratulations! Your manuscript is now being handed over to our production team.

Kind regards,

on behalf of

Dr. Stephanie S. Romanach

Academic Editor

PLOS ONE